# Learn to Play Tetris
# with Deep Reinforcement Learning

**Hanyuan Liu**
1155138650
Department of CSE
The Chinese University of Hong Kong
Shatin, Hong Kong
liuhy@cse.cuhk.edu.hk

**Lixin Liu**
1155136644
Department of CSE
The Chinese University of Hong Kong
Shatin, Hong Kong
lxliu@cse.cuhk.edu.hk

## Abstract

Tetris is one of the most popular video games ever created, perhaps in part because its difficulty makes it addictive. In this course project, we successfully trained a DQN agent in a simplified Tetris environment with state-action pruning. This simple agent is able to deal with the Tetris Problem with reasonable performance. We also applied several state of the art reinforcement learning algorithms such as Dreamer, DrQ, and Plan2Explore in the real-world Tetris game environment. We augment the Dreamer algorithm with imitation learning as Lucid Dreamer algorithm. Our experiments demonstrate that the mentioned state of art methods and their variants fail to play the original Tetris game. The complex state-action space make original Tetris a quite difficult game for non-population based reinforcement learning agents. (Video link).

## 1 Introduction

Tetris is one of the most popular video games ever created. As shown in Figure 1, the game is played on a two-dimensional game board, where pieces of 7 different shapes, called tetriminos, fall from the top gradually. The immediate objective of the game is to move and rotate falling tetriminos to form complete rows at the bottom of the board. When an entire row becomes full, the whole row is deleted and all the cells above this row will drop one line down. The game ends when there is no space at the top of the board for the next tetrimino. The final goal is to remove as many rows as possible before the game ends.

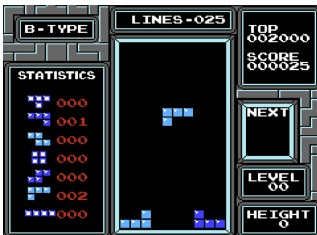

Figure 1: Screenshot of the Tetris NES with its seven types of tetriminos

Despite the simple mechanics of the game design, Tetris is a complex and difficult game. Tetris is used as an optimization benchmark [7] in which the goal is to find a policy that maximizes the number of rows removed in a game. This optimization problem contains a significantly large number of board configurations (about $2^{200}$, as the board is composed of 20 rows and 10 columns). Even if the complete sequence of tetriminos is known in advance, finding the optimal strategy is still an

NP-complete problem. This offline version of the Tetris game is known as the *Tetris problem*. It has been proven that there is a polynomial reduction between the *3-partition problem* and the *Tetris problem* [4]. So far, various methods are able to yield reasonable policies for a simplified version of the Tetris gameplay where the tetriminos are directly inserted into the game board. However, they have not reached the level of performance by human expert players playing without time pressure [1] and most of these methods fail in the real world game settings where the control sequences are explicitly required.

We choose this course project out of interest and we plan to leverage the deep reinforcement learning approach to overcome one of our childhood "nightmare": the no-win Tetris game. In this paper, we first provide a brief review related to Tetris agents. We then describe the details about the game rules of Tetris and our problem definition. After that, we present our environment settings tailored for a reduced version of Tetris game and experimental results. At last, we explore several state of the art reinforcement learning methods for real world Tetris game agent training.

## 2 Related Work

Reinforcement learning algorithms have been widely used in Tetris [7, 8, 26]. Most of these algorithms formulate Tetris as a Markov decision process and try to develop a value function of each game state. Since the state space is significantly large in Tetris, these methods use value function approximation schemes and try to optimize the value function parameters from game simulations.

Initially, hand-crafted agents had prevailed in the field until 2008 [1]. The features and evaluation functions used in these agents are mostly manually designed [5]. In the meanwhile, [26] used the cross-entropy algorithm which probes random parameter vectors in search of the linear policy that maximizes the game scores. Following [26], [27] developed the BCTS (building controllers for Tetris) using the cross-entropy algorithm with manually selected features and won the 2008 Reinforcement Learning Competition. After that, [7] proposed a classification-based policy iteration algorithm, which achieved similar performance as [27] but with considerably fewer samples.

Deep reinforcement learning has been widely applied to Atari games and achieved remarkable performance [19]. Since then we have seen some attempts in tackling the Tetris games directly with deep reinforcement learning approaches. For example, [14] trained a deep Q-network (DQN) for Tetris with temporal difference learning. [25] directly used video frames to train the agent in an end-to-end fashion. However, most of these attempts leveraged a simplified version of Tetris game without explicit control, where the pieces are directly put into the game board. In other words, they tried to solve the Tetris problem rather than play the real world Tetris game. Moreover, most of these attempts are not research-oriented and do not reach the performance of previous classic methods.

Recently, methods with latent dynamics learning and latent multi-step imagination have demonstrated reasonable performance with great sample efficiency [9, 23]. In these methods, learned world models are able to summarize the agent's experience to facilitate learning complex behaviors. Besides, data augmentation has also been popular in the context of reinforcement learning. These methods aim to improve generalization and data-efficiency through various kinds of augmentation techniques [13, 16, 17]. In terms of real-world gameplay, imitation learning techniques also demonstrate reasonable performance [3, 10, 11]. The agents that mimic experts' behavior are able to quickly adapt to a given game after behavior cloning.

We will try to solve the Tetris problem in the simplified Tetris setting used in previous works. Then we will follow the mentioned state of the art methods to develop our framework for real-world Tetris.

## 3 Problem Definition

### 3.1 Tetris Game

Tetris is a one of the most popular single-player video game among the world, which is created by Alexey Pajitnov in 1984. Players need to control pieces consisting 4 cells, called tetrominoes, and insert each tetromino into a more proper place to achieve a higher score [1, 8] .

The most famous version of Tetris contains seven different shapes of tetrominoes, looking like letters L, I, J, O, S, Z, T. When the game begins, a random shape tetromino occurs at the top of the board.

While this tetromino moves down gradually, the player can control this tetromino by rotating or moving it horizontally (left or right) until it reaches other tetrominoes' cells or the bottom of the board. Then the next tetromino, whose shape has already been given in the game, appears at the top of the board so that the player can perform a similar control. To be mentioned, the whole sequence of the shapes of tetrominoes is inaccessible except for the next one. If cells in a row are all full, their occupied space becomes free, and the score accumulates. To this end, eliminating rows as much as possible before the stack of tetrominoes reaches the top of the board is the main goal of Tetris.

## 3.2 Reinforcement Learning for Tetris

Since Tetris is a sophisticated problem, applying an advanced reinforcement learning algorithm [15, 21] in Tetris is reasonable.

**Action**. Similar to human being's control in gamepad, we define the action space as $\mathcal{A} = \{RCC, RC, L, R, D, NOP\}$, which refers to *Rotate Counterclockwise*, *Rotate Clockwise*, *Move Left*, *Move Right*, *Move Down*, *No Operation* respectively.

**State**. Although the game rules and game board settings of Tetris are a little simplistic, the naive state space is extremely large. As Figure 1 shown, formally, when we consider each tile as a binary value and the whole board as a binary image, the state size is equal to the number of distinct images, that is $|\mathcal{S}| = 2^{N \times M}$, where $N$, $M$ are the width and the height of the board respectively. The number of game states in real-world game play is much large than the number of this binary grid case, but somehow they are likely in the same order of magnitude if we take pixel colors out of consideration.

**State Reduction**. Such an extreme big state size brings training hardness in deep reinforcement learning. To construct a sample-efficient neural network and leverage the power of deep reinforcement learning, we take the reduction of state-action space into account. We will talk about more details about state-action space pruning used in the simplified version of Tetris game in section 4.1.

**Reward**. The reward function of time step $t$ formulates as follows,

$$Reward_t = \begin{cases} f(L) + g(Board) + Reward_{survival} & t < T \\ Reward_{gameover} & \text{otherwise} \end{cases} \tag{1}$$

where $T$ denotes the terminating time step, $L$ denotes the number of lines cleared at time step $t$, $Board$ is a matrix that represent the current game board, $Reward_{survival}$ and $Reward_{gameover}$ are constants, $f\colon R \to R$ is a function applied on the $L$, and $g\colon R_{m \times n} \to R$ is a function applied on the $Board$. Note that the reward function could be different in different environment.

# 4 Learn to Play Tetris

## 4.1 Environment Setup

In our experiments, we adopted a simplified Tetris simulation environment based on [20] and we will explore the mentioned state of art methods with the Tetris on The Nintendo Entertainment System (NES) using an OpenAI Gym environment developed by [12].

### 4.1.1 Simplified Environment with State-action Pruning

In the original Tetris game setting, the one-step action, i.e., movement, rotation, or no-operation, may not directly influence the game board configuration and the reward of a gameplay episode. The game board configuration only changes when a falling tetriminos becomes stationary. In other words, given a game board configuration $Board_t$ and a falling tetriminos $Piece_t$ at time step $t$, no matter what action sequences $ActSeq_i$ are taken, there are only a finite and limited number of game board configurations $Board_{t+k_i}$ when the falling tetromino is fixed on the game board at time step $t + k_i$, as shown in Figure 2a. Here we denote the transition from $Board_t$ to $Board_{t+ki}$ as a **round**. Formally, there exists at most $(MaxPieceHeight(Borad_t)/2 + 1) \times Width(Board_t) \times RotationTypes(Piece_t)$ possible next-round game configurations for $Board_t$ and $Piece_t$.

In that case, we can adopt a simplified environment setting, in which the actions are obtained from the Tetris game simulaiton on-the-fly. In each round of the Tetris game play, the states for all possible

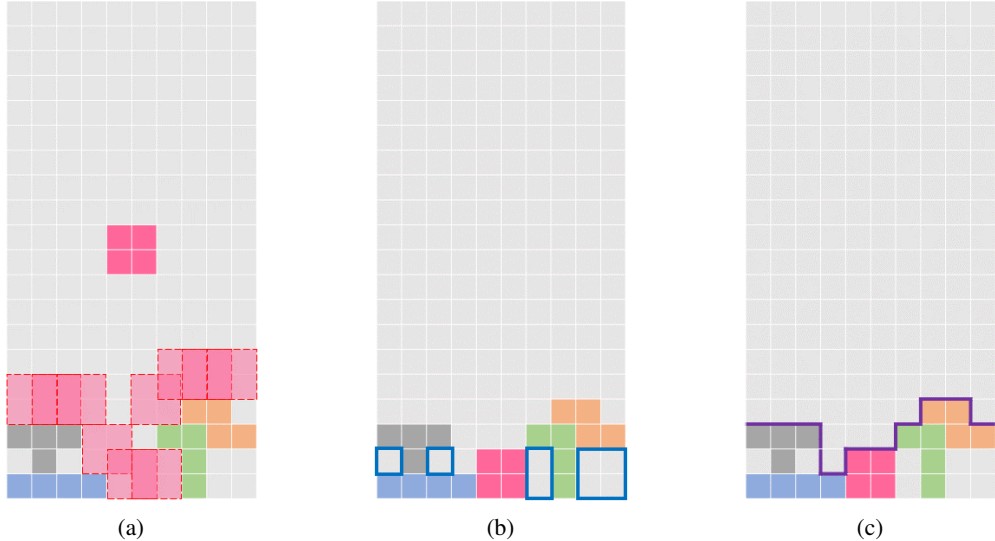

(a)                   (b)                   (c)

Figure 2: Game board configurations. (a) Pink squares: all possible game board configurations at the next game round. (b) Blue boxes: holes in the current game board configuration. (c) Purple line: the upper boundary of the current game board configuration

next-round game board configuration are collected from the environment:

$$\{state'_0, state'_1, \ldots, state'_k\} = Env(state) \tag{2}$$

Here the $state$ is a manually crafted vector that represent the game board configuration and falling piece information of one round in Tetris game. The state vector is designed as follows:

$$state = concat(\{Holes\}, \{Boundaries\}, \{FallingPiece\}, \{ClearedLines\}) \tag{3}$$

where $\{Holes\}$ denotes the number of empty cells below occupied cells of each columns in the game board, and $\{Holes\} = \{1, 0, 1, 0, 0, 0, 2, 0, 2, 2\}$ in Figure 2b; $\{Boundaries\}$ denotes the maximum height of each occupied cells in each columns of the game board, and $\{Boundaries\} = \{3, 3, 3, 1, 2, 2, 3, 4, 4, 2\}$ in Figure 2c; $\{FallingPiece\}$ is the type of tetromino fallen at current round; $\{ClearedLines\}$ is the number of lines cleared in the current round.

In this simplified environment setting, in each game round, the Tetris simulator will enumerate all possible next-round game states (or game board configurations). The action becomes selecting a next-game state from $\{state'_0, state'_1, \ldots, state'_k\}$. Thus, the action space is dynamic and depends on the current game state. The reward of this environment is simply the scores obtained at the next state, which is slightly different from the Tetris benchmark setting.

In summary, we conduct a state-action pruning based on the game design of the Tetris game. The state-action space is significantly reduced in this simplified environment setting, which leads to a smooth training process and efficient convergence of the value function.

### 4.1.2 Real-world Environment

A large number of Tetris game variants have been developed in the real world. In this course project, we will only focus on Tetris on The NES, which is a standard and popular variant of the Tetris game. We will use the *gym-tetris* environment developed by [12] to conduct our experiments. Besides, a pygame-based Tetris environment [24] is also used for accelerating the training.

In this real-world environment, the game states are the rendered frames of the Tetris NES gameplay. Besides, there are two types of action space: (1) the legal combinations of NES console inputs; (2) movement of the falling tetromino. The reward of this environment can be the change in score or the change in number of lines cleared. Thus, the state-action space is much complex than the previous simplified environment setting. We will talk about more details about this environment and our implemented methods in section 5.

Table 1: Quantitative comparison with heuristic methods

| Agents | Cleared lines (averaged on 10 episodes) |
|---|---|
| Ours | 330.1 |
| Heuristic agent 1 [6] | 136.4 |
| Heuristic agent 2 [18] | 129.4 |
| Random | 0 |

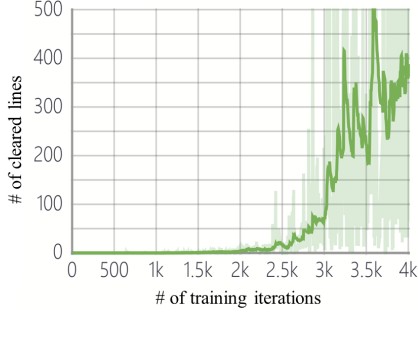 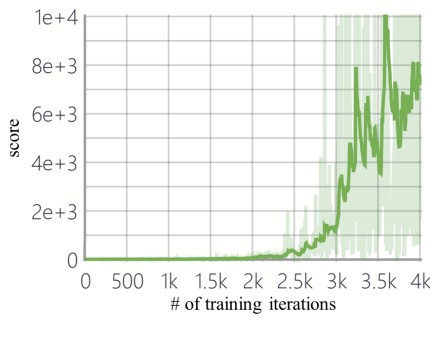

(a)  (b)

Figure 3: Simplified environment experiment results. (a) The number of clear lines. (b) Score.

## 4.2 Parameters Update

With reducing the state-action space drastically and representing the state-action pair with a hand-craft feature extractor $h(s, a)$ defined in Section 4.1.1, we denote the Q-function with a deep neural network, which takes the state-action feature as input and predicts its corresponding q-value, as $Q(h(s, a); \theta)$. To optimize the Q-function neural network parameter $\theta$, the loss is defined as

$$\mathcal{L}(\theta) = \mathbb{E}_{(s_t, a_t)}[(y - Q(h(s_t, a_t); \theta))^2] \tag{4}$$

where $y = r(s_t, a_t) + \gamma Q(h(s_{t+1}, a_{t+1}); \phi)$. where $\phi$ denotes the parameters of the target network so that the target $y$ is considered as the supervised learning target and the gradient of $y$ is ignored. The gradient of equation (4) is derived as follows,

$$\nabla_\theta \mathcal{L}(\theta) = \mathbb{E}_{(s_t, a_t)}\left[\left(r(s_t, a_t) + \gamma Q(h(s_{t+1}, a_{t+1}); \phi) - Q(h(s_t, a_t); \theta)\right) \nabla_\theta Q(h(s_t, a_t); \theta)\right] \tag{5}$$

## 4.3 Experiments

We utilized the aforementioned simplified environment to generate samples and store samples in a replay buffer, whose size is 30000. With using ReLU as the non-linear activation layer, the 2-layer multi-layer perceptron is constructed to serve as the Q-function neural network. The Adam optimizer is adopted to update the network parameter and the corresponding learning rate is 0.001. $\gamma$ in target value $y$ is set as 0.99. To enhance the model performance, the $\epsilon$-Greedy exploration technique is applied. The reward function used in this simplified environment is set to $f(L) = 100 \times L^2; g(Board) = 0;$ ; with survival bonus $Reward_{survival} = 1$ and game-over penalty $Reward_{gameover} = -2$.

As shown in Figure 3, with the increment of iterations (x-axis), our agent clears more lines and gets higher score in each game. The experiment result demonstrates that our hand-craft feature is not only lightweight but also effective under the simplified environment. We also present the quantitative comparison between the well-known Pierre Dellacherie's algorithm [6] and another heuristic algorithm used in [18, 25] in Table 1.

# 5 Extension to Real World Tetris

In order to demonstrate the hardness of Tetris game and the necessity of solution space reduction, we apply several recent advanced reinforcement learning algorithm to resolve the real-world Tetris game: NES Tetris. We also augmented the Dreamer [9] with imitation learning as the Lucid Dreamer algorithm (see Algorithm 1) and compared Lucid Dreamer with the original Dreamer, DrQ [13], and Plan2Explore [23] in the NES Tetris enviroment [12] and Pygame Tetris environment [24]. Besides, PPO [22] and DQN are served as baseline for this experiment setting.

Unfortunately, all of the mentioned methods failed in the real-world Tetris game.

## 5.1 Dreamer

Dreamer is a model-based reinforcement learning algorithm proposed in [9], which resolves long-horizon problems by latent imagination. The algorithm of dreamer contains two major parts, learning dynamics from experience and learning behavior in imagination. First, from the past experience, dreamer learns the latent dynamic with observation reconstruction and the reward prediction to represent the world model of the environment. Then, dreamer will learn behavior in imagination by walking within the latent space and predicting state values and their corresponding actions to maximize the future value predictions. The detail algorithm can be found in 1.

**Action and value models**  The action model (policy) takes the imagined state as input to predict a behavior in the imagination environment. The value models approximates the expectation of discounted imagined rewards of each imagined state $s_\tau$.

$$\text{Action model:} \quad a_\tau \sim q_\phi(a_\tau \mid s_\tau)$$
$$\text{Value model:} \quad v_\psi(s_\tau) \approx \mathrm{E}_{q(\cdot|s_\tau)}\big(\textstyle\sum_{\tau=t}^{t+H} \gamma^{\tau-t} r_\tau\big), \tag{6}$$

where $\phi, \psi$ are parameters of action model and value model respectively. With the exponentially-weighted average value $\mathrm{V}_\lambda(s_\tau)$, the learning objective of the actor and value model is given as,

$$\max_\phi \mathrm{E}_{q_\theta, q_\phi}\bigg(\sum_{\tau=t}^{t+H} \mathrm{V}_\lambda(s_\tau)\bigg), \tag{7} \qquad \min_\psi \mathrm{E}_{q_\theta, q_\phi}\bigg(\sum_{\tau=t}^{t+H} \frac{1}{2}\big\|v_\psi(s_\tau) - \mathrm{V}_\lambda(s_\tau))\big\|^2\bigg). \tag{8}$$

**Latent Dynamics Model**  The latent dynamics model of dreamer contains five critical components. The representation model encodes observations and actions to generate a continuous model states $s_t$. The observation model decode the model state to assist the training. The reward model predicts the rewards of a given model state. The transition model predicts future model states with imagination.

$$\begin{aligned}
&\text{Representation model:} & &p_{\theta_F}(s_t \mid s_{t-1}, a_{t-1}, o_t) \\
&\text{Observation model:} & &q_{\theta_D}(o_t \mid s_t) \\
&\text{Reward model:} & &q_{\theta_R}(r_t \mid s_t) \\
&\text{Transition model:} & &q_{\theta_T}(s_t \mid s_{t-1}, a_{t-1})
\end{aligned} \tag{9}$$

where $p$ denotes the distributions generating the samples in the real environment while $q$ for their approximations that enable latent imagination, $\theta$ denotes the model parameters. The loss of Dreamer is formulated as follows.

$$\mathcal{L}_{\text{REC}} \doteq \mathrm{E}_p\bigg(\sum_t \big(\mathcal{L}_{\text{O}}^t + \mathcal{L}_{\text{R}}^t + \mathcal{L}_{\text{D}}^t\big)\bigg) + \text{const} \qquad \mathcal{L}_{\text{O}}^t \doteq \ln q_{\theta_D}(o_t|s_t)$$
$$\mathcal{L}_{\text{R}}^t \doteq \ln q_{\theta_R}(r_t|s_t) \qquad \mathcal{L}_{\text{D}}^t \doteq -\beta \,\mathrm{KL}\big(p_{\theta_F}(s_t|s_{t-1}, a_{t-1}, o_t) \,\big\|\, q_{\theta_T}(s_t|s_{t-1}, a_{t-1})\big). \tag{10}$$

## 5.2 Lucid Dreamer

The Lucid Dreamer algorithm (see algorithm 1) is the Dreamer algorithm augmented with behavior cloning. We find that the Dreamer based agents cannot explore the game state faithfully, which makes it unable to learn the dynamics and policy for line clearing. Thus, a extra experience buffer $D_e$ generated by heuristic agent are feed in the Dreamer's pipeline. The hyperparameter $S$ is used to control the steps for behavior cloning. The cloning rate $\beta$ is used to balance the data sampling between the expert experience buffer $D_e$ and agent's experience buffer $D_r$.

---

**Algorithm 1:** Lucid Dreamer

---



Initialize dataset $\mathcal{D}_e$ with data collected by expert agents.
Initialize dataset $\mathcal{D}_r$ as empty.
Initialize neural network parameters $\theta, \phi, \psi$ randomly.
Set $T \leftarrow 0$, $\beta \leftarrow 1.0$.
**while** *not converged* **do**
   **for** *update step $c = 1..C$* **do**

      // Dynamics learning
      Draw $\beta B$ sequences $\{(a_t, o_t, r_t)\}_{t=k}^{k+L} \sim \mathcal{D}_e$.
      Draw $(1 - \beta)B$ sequences $\{(a_t, o_t, r_t)\}_{t=k}^{k+L} \sim \mathcal{D}_r$.
      Compute model states $s_t \sim p_\theta(s_t \mid s_{t-1}, a_{t-1}, o_t)$.
      Update $\theta$ using representation learning.
      **if** $T < S$ **then**

         // Behavior cloning
         Predict actions $a'_t \sim q_\phi(a'_t \mid s_t)$.
         Minimize the difference between $(a_t, a'_t)$.

      **else**

         // Behavior learning
         Imagine trajectories $\{(s_\tau, a_\tau)\}_{\tau=t}^{t+H}$ from each $s_t$.
         Predict rewards $\mathrm{E}\left(q_\theta(r_\tau \mid s_\tau)\right)$ and values $v_\psi(s_\tau)$.

         Compute value estimates $\mathrm{V}_\lambda(s_\tau)$.
         Update $\phi \leftarrow \phi + \alpha \nabla_\phi \sum_{\tau=t}^{t+H} \mathrm{V}_\lambda(s_\tau)$.
         Update $\psi \leftarrow \psi - \alpha \nabla_\psi \sum_{\tau=t}^{t+H} \frac{1}{2}\big\| v_\psi(s_\tau) - \mathrm{V}_\lambda(s_\tau)\big\|^2$.

   // Environment interaction
   $o_1 \leftarrow$ env.reset()
   **for** *time step $t = 1..T$* **do**
      Compute $s_t \sim p_\theta(s_t \mid s_{t-1}, a_{t-1}, o_t)$ from history.
      Compute $a_t \sim q_\phi(a_t \mid s_t)$ with the action model.
      Select action with epsilon greedy.
      $r_t, o_{t+1} \leftarrow$ env.step($a_t$).
   Add experience to dataset $\mathcal{D}_r \leftarrow \mathcal{D}_r \cup \{(o_t, a_t, r_t)_{t=1}^T\}$.
   Decay $\beta$.
   Update $T \leftarrow T + 1$.



**Model components**

| | |
|---|---|
| Representation | $p_\theta(s_t \mid s_{t\text{-}1}, a_{t\text{-}1}, o_t)$ |
| Transition | $q_\theta(s_t \mid s_{t\text{-}1}, a_{t\text{-}1})$ |
| Reward | $q_\theta(r_t \mid s_t)$ |
| Action | $q_\phi(a_t \mid s_t)$ |
| Value | $v_\psi(s_t)$ |

**Hyper parameters**

| | |
|---|---|
| Total episodes | $T$ |
| Cloning episodes | $S$ |
| Collect interval | $C$ |
| Batch size | $B$ |
| Sequence length | $L$ |
| Imagination horizon | $H$ |
| Learning rate | $\alpha$ |
| Cloning rate | $\beta$ |

Table 2: The Speed Table of Dropping Tetrominos in NES Tetris

| Level | Frames/drop | Period (sec/drop) |
|---|---|---|
| 0 | 48 | 0.799 |
| 19–28 | 2 | 0.033 |
| 29+ | 1 | 0.017 |

## 5.3 Experiment Settings

The NES Tetris environment treats each frame as a step. As table 2 suggest, directly training the agent in this environment is unacceptably slow as the tetromino would drop in every 48 steps. We hacked the Rom of the NES Tetris and fix the start level as level 19 to deal with this issue. Moreover, we applied a frameskip scheme to avoid the Delayed Auto Shift (DAS) [2] in the NES Tetris.

The frames of NES Tetris contain redundant information. Thus, we crop the frames and use a binarized image as the observation. However, we find that the agents are unable to learn clearing

Table 3: The accumulated number of cleared lines (environment with same seeds)

| Method | Total step | #cleared lines in total |
|---|---|---|
| DQN | 3 000 000 | 1163 |
| PPO [22] | 10 000 000 | 692 |
| DrQ [13] | 3 000 000 | 931 |
| Dreamer [9] | 300 000 | 55 |
| Plan2xplore [23] | 300 000 | 72 |
| LucidDreamer | 300 000 | 107 |

lines in this setup. To facilitate the training, we finally choose to use the a $20 \times 10$ martix as the environment observation.

Initially, we used the original rewards provided by the environment: (1) game scores; (2) cleared lines. However, we find that the rewards are quite sparse in the real-world Tetris environment. Thus, the rewards are set as $f(L) = 10 * L; Reward_{survival} = 0.5; Reward_{gameover} = -20$.

### 5.4 Results

The mentioned agents are unable to play Tetris after several days' training. With the given rewards, these agents learned to stack the tetromino compactly. Without enough line clearing transitions, the agents are unaware of line clearing strategies. Interestingly, the Dreamer based algorithms seems to learn some complex behavior like *Slides and Spins* in the gameplay (we will demonstrate this part in the video).

The accumulated number of cleared lines for each algorithms in a training session is shown as table 3 (Note that these numbers are only used to demonstrate the failure of all agents, the numbers are not comparable in the current setting). We can find that all the mentioned agents fails to play the Tetris. In the separated testing session, all agents are unable to clear one line in the game board.

### 5.5 Discussion

The main issue of the original Tetris is its complex state-action space. Given a tetromino and a target location, there exist a significantly large number of paths for the tetromino to reach the target location. As a result, the game state would become noisy and variational as actions are feed into the environment, which makes the random explorations unlikely to clear the lines in the game play. By contrast, the state number of the simplified Tetris environment is significantly reduced by state-action pruning, which makes it solvable by non-population based reinforcement learning agents.

## 6 Conclusion

In this course project, we successfully applied the DQN to the *Tetris Problem*. With state-action pruning, our agent is able to outperform several well-known heuristic algorithms in the simplified Tetris environment. However, we failed to train an agent that is able to play the real-world Tetris with the original state-action space like the NES Tetris. To our best knowledge, there is no non-population based reinforcement learning agent that is able to play the original NES Tetris without action space searching, state enumeration, or explicit step-wise reward shaping [1, 25]. The random exploration is unsuitable for the complex state-action space in the Tetris game. Simply using random exploration is infeasible to find the stated with lines cleared. The real-world Tetris is indeed a hard game.

In the revision phase, we will try to augment our algorithm with guided exploration, game state retraction, and action space reduction. Hopefully, these tricks would improve the mentioned algorithm.

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
