# OpenReview forum: "Learn to Play Tetris with Deep Reinforcement Learning"
_CUHK.edu.hk/2021/Course/IERG5350_

### Official Review · AnonReviewer2 · 2020-12-17
**A sound implementation of a Tetris agent trained by DQN**

**Rating:** 8
**Confidence:** 4

**Review:**

Summary:
This work resolves the Tetris problem using in a simplified Tetris environment using DQN and state-action pruning. To construct a current state vector, the authors encode the state based on the number and location of holes, boundaries, cleared lines, and the type of falling piece. Then, the agent computes all possible states in the next-round game board and finally concatenating all the possible states together as the state vector of the current state. This implementation has achieved a sound performance which significantly outperformed results presented by two heuristic agents.

On the other hand, the authors have spent effort on implementing agents trained by several state-of-the-art reinforcement algorithms. However, none of them can even equalize the results of the implementation in DQN.


Recommendation:
This work clearly shows the effort of the authors as the amounts of experiments conducted in this project for both implementations on the baseline DQN and comparison with the state-of-the-art works. A minor suggestion is it may be good to compare the results in different hyperparameter settings (e.g., DQN model different architectures) and also to give access to the project codes.

---

### Official Review · AnonReviewer3 · 2020-12-17
**Enjoyed reading this paper. I was convinced that the original Tetris is hard and your efforts are impressive.**

**Rating:** 9
**Confidence:** 4

**Review:**

## Summary:
Authors of this paper successfully trained a DQN agent to play the simplified Tetris and explored state-of-the-art reinforcement learning algorithms on the real-world Tetris.

## Strength:
* Applied effective state-action pruning on simplified Tetris and achieved impressive results.
* A lot of effort has been devoted to the explorion of various state-of-the-art algorithms on real-world Tetris. The authors even proposed the Lucid Dreamer algorithm.
* The writing is very clear and easy to follow. I like the way you illustrate with examples in 4.1.1.

## Weakness:
* I would like to see more details on the evaluation of the simplified Tetris with state-aciton pruning. E.g., ablation studies on state without _Holes_, and comparison with different hyperparameters.
* Have you tried applying state-of-the-art algorithms (e.g., Dreamer) on the simplified Tetris? Why not? I am curious on whether they can outperform DQN.

## Comments:
1. The state-action pruning in 4.1.1 looks thoughtfully-designed and impressive. Is it original or based on some ideas from existing works? Please make it clear in the paper.
2. Two sub-figures of Figure 3 looks exactly the same. Consider using only one as score is propotional to the number of cleared lines.
3. You mentioned that Tetris cannot be solved by any non-population based RL currently. Can you briefly introduce population based RL and their performance on Tetris in the paper as related work?

---

### Official Review · AnonReviewer1 · 2020-12-18
**Nice experiments and clear report**

**Rating:** 8
**Confidence:** 3

**Review:**

This report talks about a project on training a good player agent in the Tetris game using reinforcement learning algorithms. The authors show their efforts and results in training an agent to play Tetris in both a simulation environment and a real-world configuration. Their courage to conduct real-world experiments is impressive.

Besides, the problem this paper addressing is clearly described and well defined. The description of the algorithms they implemented to solve the problem is sufficient and precise. The structure of the report is also coherent. The authors provide a performance comparison between different RL algorithms, which is really appreciated. In general, the reviewer will recommend accepting this report after the authors addressing the following issues:

(1) In section 3.2, the action space of the reinforcement learning problem is introduced here.  It's commonly known that the angle of the rotation is 90 degrees for each rotation action. Maybe, it's better to suggest it in the description.

(2) In Figure 3, either one of the subfigures seems to be redundant since they are providing similar information. Is it necessary to provide both the plots?

(3) In section 4.2 and 5.1, the authors write the comprehensive mathematical formulas to describe the algorithm. However, the explanation of the notation is not sufficient for the reviewer to understand the detail. It's pretty natural for the authors to leave out the notation commonly used in this course and academic literature. It's still preferred to provide the explanation of those notations used in the formula to facilitate the reader's understanding of the described content.

(4) At the beginning of section 5 and the result section, the authors claim the involved algorithms failed in the Tetris game. Is this statement too strong to claim (like, it failed in your implementation with tens of efforts...)? It would be better the author summarized the result with a softer statement. Otherwise, the result can be easily challenged by other experienced RL researchers. (Since I'm still a beginner in RL, I'm not sure about whether the statement is appropriate. If the statement is commonly agreed among the research community, the authors are free to make this statement.)

In a word, this hard work is generally appreciated.